# Enhancing Routine Childhood Vaccination Uptake in the Cape Metropolitan District, South Africa: Perspectives and Recommendations from Point-of-Care Vaccinators

**DOI:** 10.3390/vaccines10030453

**Published:** 2022-03-16

**Authors:** Elizabeth O. Oduwole, Christina A. Laurenzi, Hassan Mahomed, Charles S. Wiysonge

**Affiliations:** 1Division of Health Systems and Public Health, Department of Global Health, Faculty of Medicine and Health Sciences, Stellenbosch University, Cape Town 7505, South Africa; hmahomed@sun.ac.za; 2Cochrane South Africa, South African Medical Research Council, Cape Town 7505, South Africa; charles.wiysonge@mrc.ac.za; 3Institute for Life Course Health Research, Department of Global Health, Faculty of Medicine and Health Sciences, Stellenbosch University, Cape Town 7505, South Africa; christina.laurenzi@sun.ac.za; 4Division of Epidemiology and Biostatistics, Department of Global Health, Faculty of Medicine and Health Sciences, Stellenbosch University, Cape Town 7505, South Africa

**Keywords:** vaccination, immunization, vaccination barriers, defaulters, interviews, South Africa

## Abstract

Vaccination, and particularly childhood vaccination, is widely acknowledged as one of the greatest public health interventions in history. Nevertheless, challenges exist that threaten the progress of childhood vaccination in many parts of the world. We investigated challenges to vaccination experienced by point-of-care vaccinators in the Cape Town Metropolitan District (Cape Metro), and reported on their current and proposed efforts to combat these challenges. Semi-structured interviews were conducted with 19 vaccinators in 16 purposively selected healthcare facilities in the Cape Metro from September to November 2019. Interviews were transcribed and analysed using thematic and narrative analysis methods. Challenges of both the demand and the supply side of routine vaccination were reported by the study participants, as were contextual challenges such as community safety issues. Defaulting was the most common challenge encountered, reported by 16 of the 19 participants. The use of computerized appointment systems, the enlistment of community health workers to track down defaulters, and the use of certificates to incentivize caregivers are part of the creative ways of mitigating some of the challenges encountered by these vaccinators. Their insightful recommendations can positively influence the landscape of childhood vaccination uptake in the Cape Metro and beyond if adapted and applied.

## 1. Introduction

The outbreak of the COVID-19 pandemic has exacerbated many issues in fragile health systems, particularly in low- and middle-income countries (LMICs) [1], and has brought vaccination to the forefront of global conversation. While routine childhood vaccinations have lagged in the last two years due to the effect of the pandemic [2], a number of persistent underlying challenges have long led to uneven coverage of childhood vaccinations, especially in LMIC settings. 

The gains of vaccination in South Africa have been well documented in literature [3,4,5]. Such gains include, but are not limited to; the elimination of smallpox, significant reduction in child mortality due to vaccine preventable diseases (VPDs) such as diphtheria and pertussis, and near elimination of VPDs like measles and wild polio virus. However, in South Africa, high levels of inequality and health system underperformance have contributed to poor childhood vaccination coverage [5,6,7,8]. Specific pragmatic challenges have included missed opportunities for vaccination [7], lack of awareness of vaccination schedule by primary caregivers [9], and consistent vaccine stock-outs [10]. Vaccine hesitancy, defined as the delay in acceptance or refusal of vaccines despite the availability of vaccination services [11], has also been identified as a challenge to vaccination in the country [12]. 

While government and other stakeholders such as civil societies, charity organizations and non-governmental organizations (NGO) have attempted to counter these challenges, the ongoing COVID-19 pandemic has interrupted and aggravated the challenges of routine vaccination programs in many places including South Africa [8,13,14,15,16]. As such, investigating challenges and existing solutions to achieving optimal vaccination coverage and uptake of routine childhood vaccination is of vital importance. Wiysonge et al. in 2012 documented challenges identified by managers within the Expanded Program on Immunization (EPI), a national program aimed at improving childhood immunization and the various solutions that they proffered to tackle them [5]. Training and supportive supervision of healthcare workers, regular audits, and feedback of relevant healthcare processes and systems were among the evidence–based, tenable solutions that were offered [5]. 

Nevertheless, there is limited evidence documenting the creative approaches that point-of-care vaccinators, who directly experience these challenges, are using. These individuals, whose perspectives are often omitted from the literature, may be able to provide applicable solutions and recommendations given their frontline experiences. This advantageous position of constantly encountering these challenges enables them to gain valuable insights, and with the benefit of hindsight, proffer plausible solutions. 

This study thus investigated what the point-of-care vaccinators are offering in the bid to continue to improve vaccination deliverables in their work-community. Conducted in the Cape Metropolitan District of South Africa in the period prior to the outbreak of the COVID-19 pandemic, this study documents the efforts and recommendations of the participants at mitigating challenges and increasing vaccination uptake and coverage that could be implemented in the immediate region and other similar settings. 

## 2. Methods

### 2.1. Study Design, Setting and Study Sites

This exploratory qualitative study was conducted in the Cape Metropolitan District (referred to as the Cape Metro area) between September–November 2019, in 16 purposively selected healthcare facilities. To select facilities, we reviewed facility-linked electronic data on childhood vaccination to identify the number of fully-vaccinated children under one year from April 2017–March 2018. Facilities with higher numbers were chosen to maximize chances of connecting to vaccinators with substantial experience. We obtained permission to access the facilities and interview participants from the National Health Research Database, the Western Cape Department of Health, and from City Health, City of Cape Town.

The Cape Metro area has an estimated population of 4.18 million (2019) [17], which comprises approximately 61% of the total population of the Western Cape Province in South Africa. The 16 facilities were located in peri-urban areas of the Cape Metropolitan District. At least one facility was included from each of the eight health sub-districts of the Metro [18]. 

### 2.2. Study Population and Sample Size

The study participants were point-of-care vaccinators with five years or more post-qualification experience. Vaccinators with a minimum of 5 years post qualification experience were judged to be “rich cases” whose years of experience would have exposed them to many of the challenges of routine childhood immunization, and enabled them to be able to garner insight to develop and implement plausible solutions to mitigate these challenges. In selected facilities where current vaccinators had less than the requisite years of experience, other suitable health officials with previous vaccination experience, such as facility managers, were included in the study population. 

### 2.3. Data Collection and Analysis

The principal investigator (EO) conducted key informant interviews (KIIs) with the vaccinators in person, and all interviews were conducted in English. After the initial introductory session, participants were given a summary of the study and their potential role, and informed consent was obtained. The semi-structured interview guide, designed specifically for the study, was used to structure the KIIs. Open-ended questions were used to explore issues of interest, including challenges in achieving optimal routine childhood vaccine coverage and uptake in the population served by these vaccinators, participants’ efforts at mitigating these challenges, and suggestions and recommendations to further enhance vaccination uptake and coverage in the Cape Metro area. 

Audio files and verbatim transcriptions of the interviews were imported into ATLAS.ti qualitative data analysis software (version 8.4.24. Scientific software development, GmbH Berlin, Germany). Hard copies of transcripts were broadly coded after the initial reading. These codes were used as guides for the subsequent coding of the electronic copies for the issues of interest detailed above. The codes were grouped together based on the related issues and were organized into a thematic map (network). We used thematic and narrative analysis methods [19,20,21] to elucidate, present, and discuss the findings of the study. Participants’ direct quotes were included where appropriate to substantiate the issues raised. 

### 2.4. Ethics Approval 

The study received ethics approval from the University of Stellenbosch’s Health Research Ethics Committee, Reference # S19/01/014 (PhD). The study also obtained approval from the Western Cape Department of Health after application approval by the National Health Research Database (reference number WC 201906 015) available at: https://nhrd.hst.org.za/Proposal (accessed on 1 September 2019), and from City Health, City of Cape Town (Reference number 24480; Project ID 8160) available at: http://web1.capetown.gov.za/web1/Mars/ProjectAttachment/Read/0/8160 (accessed on 1 September 2019).

## 3. Results

Nineteen point-of-care vaccinators including registered nurses (*n* = 9) and enrolled nurses (*n* = 10) were interviewed. Enrolled nurses work under the supervision of registered nurses, the former generally have about two years of a different nursing curriculum education compared to the four years of the latter. Interviews lasted fifteen minutes on average. This section presents the challenges raised, in brief, as well as the solutions that vaccinators shared. 

### 3.1. Reported Challenges and Barriers to Optimal Vaccination Coverage in the Cape Metro

The challenges reported by the interviewed vaccinators included those affecting the primary caregivers as perceived by the vaccinators (demand-side challenges), and those directly affecting the vaccinators themselves (supply-side challenges). A number of contextual barriers raised affected both caregivers and vaccinators. Table 1 contains a summary of these challenges, and illustrative quotes from participants. It is important to note that mothers or other primary caregivers were not interviewed as this was not in the scope of the study. 

Defaulting, described as the missing of scheduled vaccination for any reason including health facility problems such as cancelled sessions or vaccine stock-outs [22,23], emerged as the most commonly reported demand-side challenge. Defaulting was attributed to various reasons caused mainly by economic factors, and reflective of the clients’ socio-economic status. 

### 3.2. Efforts to Mitigate Challenges Encountered

Participants shared different ways that they were attempting to mitigate the challenges encountered in the course of their duties. They also suggested some valuable recommendations that they felt could improve vaccination uptake and coverage in the Cape Metro area. 

#### 3.2.1. Efforts Directed at Demand-Side Challenges

Most of the challenges reported by participants related to barriers impeding primary caregivers’ ability to access vaccination services. Most of the existing and proposed solutions were geared towards addressing defaulting. The use of a well-documented, computerized, appointment, reminder, and recall telephonic system was described as a successful and efficient way to trace, remind, and engage defaulters. These systems were successfully deployed in many of the facilities in the Cape Metro. Participant C1.D1 described:


*“What we did now in the clinic or so previously, we let everyone come and they wait three, four, five hours but now we have an appointment system in place; 8 o’clock, 10 o’clock, 12 o’clock and 2 o’clock and it works for them”. Explaining further, she said… “It works for us, then the mothers don’t sit so long. You don’t get this moaning and groaning too much anymore. Because I know my appointment is for 12 o’clock, so I come to the clinic at 11:30. I sit for an hour and a half, and then I go. Where [as] I used to come at 7:30 at the gate, then I leave at 1 o’clock. So we are having an appointment system in place and that is also helping with our immunizations.”*


The effect of a well-managed reminder and recall telephonic system was similarly alluded to by participant C3.D2.


*“We have computers now. We book them on the computer and then we write on their cards. … The computer is reminding them the day before the appointment. Even when you are still busy with them here, after you have checked them in on the computer, the message goes through to their phone. And then also reminds them a day before, about their appointment.”*


This appointment system enabled caregivers to feel more confident in the services provided, with a net positive impact on immunization in the area. Moreover, as another participant suggested, children easily become fussy and fretful in overcrowded facilities, further contributing to caregiver frustration and reluctance to return for routine immunizations, *“So again, I think that is the issue and whether it’s a sense of the younger children are more difficult to deal with in a crowded facility that’s why they don’t come on time that might play into it…”* Participant C5.1. D1. 

Participants described additional tracking processes through the use of community health workers (CHWs), also called community care workers (CCWs), to trace, locate, and engage defaulting care givers, creating a sense of shared responsibility between vaccinators and CCWs, *“But with the other mothers, yes, they do miss their appointments but when we recall them because we have CCWs who are recalling them for us. So then they will come to the clinic”* (Participant C6.1D2). 

Collaborating with non-governmental organisations (NGOs) provided another effective way to combat defaulting. One participant indicated that collaborating with NGOs afforded them the opportunity to offer another chance for vaccination to clients who would have otherwise defaulted or missed out completely on their scheduled vaccination, *“The only thing is the mothers are scared to come on this side, it’s because of that* [gang activity]. *Nowadays we have a clinic that goes to* [NGO name]. *We take the service to them* [there], *and there they will get immunised”* (Participant C1.D1). In a post interview discussion, this participant explained the modalities of this collaboration further. The clients are informed of specific fixed days in the month which vaccinators are stationed at the premises of that particular NGO which is located in a ‘safe’ area in the vicinity of the facility. The mothers are thus encouraged to bring their children for vaccination on these fixed days at the premises of the NGO. 

Participants further described the use of certificates and promotion as incentives for caregivers. Describing this innovative effort, participant C8.D2 said…


*“For me alone, I am doing what you call it: a certificate of the children at 18 months. I try not to miss it. When I reach to 18 months, only in [facility name], I give them the certificate, to tell them that you reached 18 months, you did a good job. Now I am discharging you from my room, you are going to room 4 next time for 2 years. They like it and everyone, they are so motivated.”*


This same vaccinator provides additional layered social assistance to caregivers, which serves dual purposes of the proper child documentation necessary for accessing essential needs, and identification of the bona fide mother or legal guardian of the child by accompanying the mothers or primary care givers to the police station to swear affidavits that they are either the bona fide mothers or legal guardians, or to give the proper surnames to the babies. Through offering this assistance with the correct identification and documentation of a child by sworn affidavit, this vaccinator endears herself to the caregivers, and they in turn are motivated to comply with the timeous immunization of their children.


*“…And what we did again in [facility name], we went to the police station to discuss with them the affidavit; [for example] if the child is coming from the country [from a rural area to the city], we must get the name of the person [that] will be responsible for the child [a sworn affidavit of ownership or guardianship of the child]. The person we can reach.” And, in this area they know, all of my children, they come to my room. They are already changed [that is, they now have the correct names preferred by their parents or legal guardians] and I changed them all. Then I give them that thingy [affidavit of correct identification], that is going to go with the card of the baby when they go to school.”*
(Participant C8.D2)

#### 3.2.2. Efforts Directed at Supply-Side Challenges

Participants also shared some solutions that eased barriers to provision of vaccination services. One such solution was enlisting expanded public works programme (EPWP) beneficiaries, also called ‘queue marshals’, to assist at the facilities. The EPWP is a job creation initiative of the Western Cape Province. *“You know what, every six months they appoint the EPWP people, do you know the EPWP programme it’s where the city helps get people out of the* [poor] *circumstances they stay in, then they work for six months in the clinic.”* (Participant C9.D1). One important caveat was that EPWP candidates enlisted often lacked basic training, hampering the impact that these queue marshals have, and she suggests basic training for them to optimize their impact in the clinics:


*“Then there is a queue marshal. The queue marshal doesn’t have any clinic experience, now she is coming to work in the clinic. Her job is to educate but she herself knows nothing… if they could get trained queue marshals, that will also help us. Because then she can do the health education.”*
(Participant C9.D1)

This same participant expressed similar sentiments about another on-going intervention to alleviate some of the supply-side challenges. She opined that the re-training of ‘agency nurses’ (nurses employed and deployed by employment agencies) to familiarize them with the routine operations of the facility before they are allowed to assume important responsibilities will go a long way to ease the burden of time constraint on other staff members. She lamented having to cross-check and sometimes redo specific tasks assigned to these agency nurses, which adds to the existing pressure of time constraints. As she describes *“And then they work with the agency sisters which some of them don’t even know what’s happening in the clinic. You understand, then one of us needs to audit her work and there’s no time to do that…”* (Participant C9.D1)

#### 3.2.3. Structural Suggestions and Recommendations to Improve Vaccination Coverage 

Participants provided a number of other recommendations that are structural in nature, including the expansion and simplification of the expanded programme on immunization (EPI) charts to include information about the advantages of vaccination, as well as the possible repercussions of delaying or missing routine immunizations on schedule. Participant C5.1.D1 explains that… 

*“The EPI is a wonderful poster* [but] *it doesn’t explain a little bit more in-depth. …If that EPI* [chart] *could be expanded to add on another column [which explains the purpose/or advantage of each vaccine], [and] if that can be placed at point of care, just above where the immunisations get given, that would sort of help a lot in terms of credibility* [and] *in terms of me explaining…”*

Engaging mass media establishments such as television, radio, and outdoor billboard advertising was another recommendation proffered by one participant. Participant C5.2D1 observed that, in contrast to the usual including staff-driven vaccination outreach campaigns that prompts the uptake of the seasonal flu vaccination, a local soap opera had prompted many residents of her community to turn up en masse in 2019. This series had featured a number of scenes in which the Zika virus epidemic was mentioned, prompting residents of the community to think that the flu vaccination was vaccination against the Zika virus disease. Therefore, they came out voluntarily in their numbers to be vaccinated. This participant suggested that the power of mass media could be harnessed to promote routine childhood immunization, encourage positive perception and reception by caregivers, and improve vaccination coverage overall, *“So maybe if we can maybe* [mentions one of the popular indigenous soap operas]—*this is just a suggestion, incorporate someone with a baby going for immunisation to a health facility. And there the explanation can take place; we will reach a lot of people through that.”* (Participant C5.2D1)

Having ‘gazebo days’ was another useful recommendation. One participant suggested that healthcare workers could set up a gazebo linked to the Department of Health at the same sites where state-provided child support grants are given out. Because caregivers usually bring child recipients to the grant collection stations, healthcare workers could use the opportunity to provide catch-up immunizations and other basic medical services for eligible children. *“The children who are getting support grants. Like whenever they are in the queue and then we as the Department of Health, we can have a gazebo day and have an outreach there. While they are waiting in the queue then their kids are getting the vaccines.”* (Participant C7.D1) 

Engaging employers to honour ‘proof of attendance’—documentation to explain their missing work to bring children for vaccinations—was another important suggestion to improve coverage. *“We would explain to them that we are going to give them the certificate, the proof of attendance but the employer doesn’t care about that”* (Participant C13.D2). Another participant from another facility shared this experience: 


*“The problem that I encounter, the other mothers are willing but because of their bosses. Their employers don’t give them off days to come for their immunisation. Because others they want but still when we give proof of attendance that they were here, still they don’t get paid for that day. They say they can’t give them because they are not taking that proof of attendance written by a professional nurse. It must be written by a doctor…”*
(Participant C7.D1)

This participant recommended that the government compel employers to honour proof of attendance signed by nurses, and that employers should be incentivized to allow their employees to bring their dependents for routine vaccination timeously.

Finally, restoring “return-to-school” vaccination days was an additional suggestion. One participant recalled, *“Those years, we did immunising at the schools also, go around to the schools and we didn’t miss out on children”* (Participant C2.2.D2), identifying an effective means to reach children directly and circumventing the additional challenges in engaging caregivers. 

Overall, the findings of this study highlight various challenges faced by point-of-care vaccinators in the Cape Metro, the major one being vaccination defaulting. The findings also bring together various ways employed by these vaccinators to counter the identified challenges, and viable suggestions and recommendations to improve and optimise vaccination coverage and uptake, particularly of childhood vaccines in the Cape Metropolitan District. 

## 4. Discussion

To improve routine childhood vaccination coverage in LMICs, it is essential to have a clear sense of existing challenges and embedded solutions. This study explored challenges, existing solutions, and plausible recommendations to improve routine childhood vaccination coverage in the Cape Metropolitan District of South Africa, engaging point-of-care vaccinators. These frontline vaccination service providers, who are not routinely engaged in the literature, have a distinct vantage point from which to share current solutions and offer further recommendations. 

Vaccination defaulting is the most common challenge reported by the study participants and has been previously reported in similar literature [22,23,24,25,26,27,28,29,30,31,32]. Zewdie et al., 2016 [27] who in a similar study investigated the challenges of defaulting in Hadiya zone in Southern Ethiopia, reported a lack of viable defaulter tracing system in their findings. In contrast, our study reports well-established and effective defaulter tracing systems. It is possible that differences in the populations of the two studies is responsible for this observed difference. 

Some of the reasons for defaulting reported in this current study have been previously reported in literature [22,25,26,27,28,30,31,33,34]. As many challenges shared reflect inequitable socio-economic conditions, our findings point to the need for a multi-dimensional approach to increasing and optimizing vaccination uptake and coverage. Such an approach might include increased collaboration between different departments of government such as Health and Social Services; and also with non-governmental organizations. This present study responds to the knowledge gap for improving routine childhood immunization uptake not only in the Cape Metro, but potentially also in other similar urban and peri-urban populations. 

Some of the insightful solutions used and recommended by participants, such as collaborations with NGOs to circumvent gangsterism, and assistance with the correct documentation of children, addresses context-specific challenges that were not previously reported by other studies. Similarly, others—like the use of certificates and promotion incentives, having gazebo days in conjunction with NGOs, and engaging media houses—were innovative ideas provided by participants directly. These findings in particular underscore the importance of enlisting point-of-care vaccinators, as well as other frontline providers, in efforts to promote vaccination across context and setting, involving them in generating solutions and crafting policy. 

The interplay of multi-dimensional approaches employed by individuals such as those enumerated above, and supportive systems provided by the government highlighted below will make significant differences in the vaccination deliverables in the Cape Metro area if developed and harnessed optimally. This is the main thrust of this study. 

Government-linked efforts to ameliorate some of the human resource-related challenges, such as the enlistment of EPWP beneficiaries [35], was also a context-specific solution. The provision of computerized appointment and tracking systems, though not peculiar to the Cape Metro area, is one of the most lauded interventions of the government that inspired and facilitated the multi-dimensional efforts of individual vaccinators. 

Other studies including Brown [36] and Eze [37] also found the use of computerized reminder and recall telephonic systems to be highly effective in tracking down defaulters for ‘catch up’ immunizations in resource-constrained, urban dwelling areas with good cell phone coverage. Likewise, Aregawi et al. [22] identified a positive impact of community health workers in identifying immunization defaulters in Ethiopia, improved by monthly visits. The use of computerised appointment and tracking systems and the engagement of community health workers contributes significantly to the easing of the burden on the health facilities and practitioners. These two strategies were found to increase overall uptake.

The importance of involving frontline vaccination service providers in any vaccination challenge investigation and addressing endeavour as done in this current study cannot be overemphasised. They provide valuable insights from their vantage point of first-hand experience of the situations, and also plausible solutions and recommendations. An example of such is the innovative action of rendering assistance with the correct identification and documentation of a child by a sworn affidavit at a police station by participant C8.D2. If such a service is supported by policy and widely practised, those clients who otherwise would have missed out on the routine immunization of their children or wards would have the opportunity of presenting them at the nearest health facilities in their new locations both for immunization and documentation. 

The major limitation of this study is that only vaccinators were interviewed, no mothers or primary care givers were interviewed. This may introduce some level of bias in the demand-side challenges reported by the vaccinators. However, this was done in keeping with the scope of the study. Moreover, all the facilities included in this study were government facilities located in the peri-urban areas of the Cape Metro area. No private facilities or facilities located in the more affluent area of the Metro were included in the study. The generalizability of the findings may be limited as the Cape Metro is different in population and demography to the rest of the Western Cape Province (being the only urban district, with the rest classified as rural districts). Nevertheless, our findings may prove beneficial if solutions and recommendations are considered and adapted based on context in other parts of South Africa or similar settings. 

## 5. Conclusions

This research documents the efforts and results of the point-of-care vaccinators in the Cape Metro in creatively mitigating both common and distinct challenges encountered in the course of their duties. It also reports on the recommendations proposed by these frontline healthcare workers, to improve vaccination coverage and uptake in the Metro. Our findings reiterate the importance of listening to and enlisting perspectives from frontline health providers, especially as context and setting shape the way in which childhood vaccinations can be delivered and taken up. 

To ensure that the insightful solutions and recommendations proffered by these frontline vaccination service providers are taken up for appropriate action, a copy of this article will be uploaded on the National Health Research Database and on the City Health, City of Cape Town websites. A copy of the article will also be attached to the report of the study expected to be submitted to the Western Cape Department of Health at the completion of the study. Furthermore, the findings of this study will be presented at national and international conferences and other appropriate platforms. 

Future research should include point-of-care vaccinators in private facilities and government facilities located in a more affluent area of the Cape Metro. In addition, mothers and other primary caregivers should be included in future research. 

## Figures and Tables

**Table 1 vaccines-10-00453-t001:** Challenges to achieving optimal vaccination coverage.

Demand-Side Challenges	Illustrative Quotes
**Defaulting**	*“They always bring their babies for immunization.* [Nevertheless], *you get mothers who default. Defaulters mean they don’t bring their babies* [on time for their scheduled routine immunization].” (Participant C1.D1)
Constant internal migration or relocation	*“No, no, it’s not that they don’t want vaccine. Most of those who don’t give* [their children] *the injections are those who are going to the Eastern Cape* [Province]... *and when they are there, they don’t take their children to the clinic. That is their problem or they change their address. Maybe they stay in Gugulethu. After two or three months, they go stay in Khayelitsha. After that she went to Camps Bay, so they are moving around, that is their problem.”* (Participant C10.D2)
Employment security	*“They are scared that they could lose their jobs if they come to the clinic. Their jobs are not safe at all…”* (Participant C13.D2)
Adverse weather conditions	*“Most of them are coming, depending also on the weather. If it is raining you don’t get them, you don’t get them here”.* (Participant P3.D1)
Cost of transportation	*“Sometimes they don’t have money to get here.”* (Participant P3.D1)
Lack of awareness on the importance of (timeous) immunization (motivational barrier)	*“There has been many who are late for their immunisations…and the reason for that, I found because they don’t understand the importance of immunising. They don’t understand the importance of returning within the 6 weeks, 14 weeks, coming within the correct time frames. They don’t understand that you are boosting every time every 4 weeks and 6 months for measles. So there’s a lot of misunderstanding in terms of “why I should come for immunisation?”, “why I should come on time for my immunisation?”* (Participant C5.1D1)
Long queues at the clinics	*“The other reason why the mommy doesn’t want to come to the clinic, she doesn’t want to sit in long queues at the clinic…”* (Participant C2.2.D2)
**Supply-side challenges**	
Staff shortages	*“…and the next thing is, there is no staff. There are no hands to go out to do these things… We’re really pulling ourselves because there is no staff. Staffing is a problem. Shortage of staff is one of the biggest problems…”* (Participant C1.D1)
Time constraints	*“…you know what, the staff complements, too many clients per one registered nurse, you understand. And then everything needs to be so fast, you don’t even get a proper lunch time and I’m sure my other colleagues feel the same because its pressure after pressure. You understand and we don’t make the time to really sit the client down and talk to the mother, ask the mother what are your challenges, what do you want to know? We just inject and off you go…”* (Participant C9.D1)
**Contextual challenges**	
Substance abuse	*“… But the substance abuse is our biggest problem in this area. That’s why the mother doesn’t bring the child.”* (Participant C2.D2.2)
Gang-related activities	*“…because it’s just that in our surroundings we have lots of gun shootings. There’s lots of gangsterism and drugs. The only thing is the mothers are scared to come on this side, it’s because of that…”* (Participant C1.D1)

Bold typface denote the major challenges. All verbatim quotes are in italics.

## Data Availability

All data relevant to the study are included in the article.

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
