# Peer review of "Enhancing Routine Childhood Vaccination Uptake in the Cape Metropolitan District, South Africa: Perspectives and Recommendations from Point-of-Care Vaccinators"

_vaccines, 2022, doi:10.3390/vaccines10030453_

Round 1

Reviewer 1 Report

The results are presented in a non-scientific narrative. I was expecting a compilation of the results and statistical data processing to ensure that the results of the survey will highlight the main aspects detected during the investigation and the impact of each detected event.

Author Response

Comment: The results are presented in a non-scientific narrative. I was expecting a compilation of the results and statistical data processing to ensure that the results of the survey will highlight the main aspects detected during the investigation and the impact of each detected event.

Response: We thank the reviewer for the comment.

However, the authors will like to reiterate that this is an exploratory qualitative study conducted with key informants (point-of-care vaccinators) using a semi-structured interview guide as indicated in the manuscript. The data (interview transcripts) were organized and coded in ATLAS.ti qualitative data analysis software (version 8.4.24).  Also, as indicated, thematic and narrative analysis methods (reference19–21) were used to elucidate, present and discuss the findings of the study; and participants’ direct quotes were included where appropriate to substantiate the issues raised.

No survey was conducted as this was not a quantitative study, hence, no statistical data processing was required nor conducted, as typical of qualitative studies.

Reviewer 2 Report

I highly value the approach taken and the focus on point-of-care vaccinators in an attempt to enhance routine childhood vaccination, using a qualitative approach. This resulted in an interesting, creative document that is useful, also ‘outside’ the region.

There is a semi-structured interview guide, but this guide is not provided in the document, there is perhaps value to add this document as a supplement ? Can you elaborate somewhat more on how this guide has been constructed and ‘validated’

To what extent could answers be impacted by COVID ? Can the authors add some information on the timing of the interviews versus the covid ‘waves’ ?

Table 2: something went wrong in the editing, and the challenges do not match with illustrative quotes.

Author Response

The authors use this medium to thank this Reviewer for the efforts at reviewing our manuscript and the constructive feedback given to help to further improve the quality of the manuscript.

Please find below detailed response to each point raised.

Comments and Suggestions for Authors

Point 1: I highly value the approach taken and the focus on point-of-care vaccinators in an attempt to enhance routine childhood vaccination, using a qualitative approach. This resulted in an interesting, creative document that is useful, also ‘outside’ the region.

Response 1: The kind comments and recommendation of this Reviewer is well appreciated

Point 2: There is a semi-structured interview guide, but this guide is not provided in the document, there is perhaps value to add this document as a supplement? Can you elaborate somewhat more on how this guide has been constructed and ‘validated’

Response 2: The interview guide was constructed by the study team which included two experts in the field of vaccines and vaccination, with input of two international experts, one in the field qualitative studies and the other in mixed methods. The interview guide was focused on vaccine hesitancy, which is the overarching objective of the broader study of which this study is a part. The eagerness and willingness of the vaccinators to talk about other challenges that they are encountering and their innovative ways of mitigating them spurred the PI to explore this unscripted area of interest. Therefore, including the interview guide in this article may not add any more value to it.  The interview guide was not formally validated but was based on expert input as indicated above.

Point 3: To what extent could answers be impacted by COVID? Can the authors add some information on the timing of the interviews versus the COVID ‘waves’?

Response 3: As indicated in the introduction and methods section of the manuscript, the interviews were conducted prior to the outbreak of the COVID-19 pandemic, between September-November 2019. Therefore, the answers were not impacted by COVID-19 waves. The relevant statements have been highlighted in green in the manuscript for greater visibility to the reviewer.

Point 4: Table 2: something went wrong in the editing, and the challenges do not match with illustrative quotes.

Response 4: We agree that the alignment of the challenges to their corresponding illustrative quotes in Table 2 has been affected perhaps due to the recommended Table format of the journal.

This has been addressed in the revised manuscript. Moreover, the heading has been changed to ‘Table1’.         

 Also, the Table in Word document (though in a format not stipulated by the journal) has been attached for consideration as a replacement in the published manuscript to prevent the misalignment of the challenges to their corresponding illustrative quotes. This will also be

Reviewer 3 Report

This paper seeks to highlight solutions used by vaccinators to improve vaccine uptake.  The paper is a qualitative study detailed the ideas.  Major considerations of the paper include

  1. It would be nice to have a survey of the 19 vaccinators with a likert scale indicating which problems they perceive as the most pressing. Then you could compare the ideas and if they are addressing the perceived major problems.
  2. The selection of only vaccinators with 5 years of experience needs justification.  
  3. Table 2 formatting is confusing as the rows dont line up and so it takes considerable work to match them.
  4. Adding a table of the good ideas would be helpful.
  5. A review of other studies and if they matched ideas would be helpful.
  6. A discussion of how to dispurse the ideas and information would add to the weight of the paper.
  7. Adding an analysis of how much uptake could improve with the two strategies in paragraph (line 337)would be helpful.
  8. A paragraph on supply-side challenges is missing.

Minor

  1. Breaking the large paragraph from introduction at 48 and discuss government /ngo would be helpful.
  2. Move "interviewed lasted fifteen minutes on average to results section 3.

Author Response

The authors use this medium to thank this Reviewer for the efforts at reviewing our manuscript and the constructive feedback given to help to further improve the quality of the manuscript.

Please find below detailed response to each point raised.

Comments and Suggestions for Authors

This paper seeks to highlight solutions used by vaccinators to improve vaccine uptake.  The paper is a qualitative study detailed the ideas.  Major considerations of the paper include

Point 1: It would be nice to have a survey of the 19 vaccinators with a likert scale indicating which problems they perceive as the most pressing. Then you could compare the ideas and if they are addressing the perceived major problems.

Response1:

Your proposal is noted and will be discussed with the research team as a possible follow up study.

Point 2: The selection of only vaccinators with 5 years of experience needs justification. 

Response 2: The justification for the minimum of 5 years post qualification experience required of the interviewed vaccinators is as follows:

Vaccinators with minimum of 5 years post qualification experience were judged to be “rich cases” whose years of experience would have exposed them to many of the challenges of routine childhood immunization, and enabled them to be able to garner insight to develop and implement plausible solutions to mitigate these challenges.  This justification is now included and highlighted in blue in the methods section of the manuscript.

Point 3: Table 2 formatting is confusing as the rows dont line up and so it takes considerable work to match them.

Response 3:

We agree that the alignment of the challenges to their corresponding illustrative quotes in Table 2 has been affected perhaps due to the recommended Table format of the journal.

This has been addressed in the revised manuscript. Moreover, the heading has been changed to ‘Table1’.

Also, the Table in Word document (though in a format not stipulated by the journal) has been attached for consideration as a replacement in the published manuscript to prevent the misalignment of the challenges to their corresponding illustrative quotes. This will also be pointed out in the cover letter to the Editor.

Point 4: Adding a table of the good ideas would be helpful.

Response 4: Adding a table of the good ideas might be helpful, but considering the challenges being experienced by the inclusion of the current Table 1, in our opinion, adding another table that is not an absolute necessity might be counterproductive. Moreover, as the good ideas are clearly described and discussed in the manuscript, putting them again in a table format will be repetition. It will also increase the length of the article, which may be discouraging to some readers.    

Point 5: A review of other studies and if they matched ideas would be helpful.

Response 5: A lot of the suggestions and recommendations given by the study participants are original and unique, and were not found reported in available literature. Those that were found reported elsewhere were briefly described, compared and contrasted as appropriate in the discussion section.

Point 6: A discussion of how to disperse the ideas and information would add to the weight of the paper.

Response 6: A short paragraph indicating how the ideas and information would be dispersed especially to the concerned health governing authorities in the immediate study environment has been included in the conclusion section of the manuscript. It is highlighted in blue.

Point 7: Adding an analysis of how much uptake could improve with the two strategies in paragraph (line 337) would be helpful.

Response 7: Appropriate data required to conduct an analysis of how much routine childhood vaccination uptake could improve with the two strategies of computerised appointment and tracking systems and the engagement of community health workers is not available to the research team. Nevertheless, their positive effects was alluded to by most of our study participants,  that these eases the burden on the facilities and the health care providers as indicated in the discussion section of the manuscript, and also in the cited literature.

Point 8: A paragraph on supply-side challenges is missing.

Response 8: Unlike the demand-side challenges that were relatively many necessitating a brief introductory paragraph, only two supply-side challenges were identified from the responses of the participants. The two are staff shortages and time constraints. These are included in Table 1 of the revised manuscript, and the efforts at mitigating them reported in section 3.2.2 of the findings section.

Minor

Point 9: Breaking the large paragraph from introduction at 48 and discuss government /ngo would be helpful.

Response 9: A paragraph break has been effected in the revised manuscript as recommended.

Point 10: Move "interviewed lasted fifteen minutes on average to results section 3.

Response 10: The suggested recommendation has been effected in the revised manuscript.

Round 2

Reviewer 1 Report

It is OK for publication